# Photosensitizer-free visible-light-promoted glycosylation enabled by 2-glycosyloxy tropone donors

Jing Zhang[1], Zhao-Xiang Luo[1], Xia Wu[1], Chen-Fei Gao[1], Peng-Yu Wang[1], Jin-Ze Chai[1], Miao Liu[1], Xin-Shan Ye [1] & De-Cai Xiong [1,2] ✉

Photochemical glycosylation has attracted considerable attention in carbohydrate chemistry. However, to the best of our knowledge, visible-light-promoted glycosylation via photoactive glycosyl donor has not been reported. In the study, we report a photosensitizer-free visible-light-mediated glycosylation approach using a photoactive 2-glycosyloxy tropone as the donor. This glycosylation reaction proceeds at ambient temperature to give a wide range of *O*-glycosides or oligosaccharides with yields up to 99%. This method is further applied in the stereoselective preparation of various functional glycosyl phosphates/phosphosaccharides, the construction of *N*-glycosides/nucleosides, and the late-stage glycosylation of natural products or pharmaceuticals on gram scales, and the iterative synthesis of hexasaccharide. The protocol features uncomplicated conditions, operational simplicity, wide substrate scope (58 examples), excellent compatibility with functional groups, scalability of products (7 examples), and high yields. It provides an efficient glycosylation method for accessing *O/N*-glycosides and glycans.

Carbohydrates are widely distributed in nature and play pivotal roles in bacterial and viral infection, intercellular recognition, immune regulation, inflammation development, cancer cell metabolism, and other physiological or pathological processes[1,2]. Carbohydrate-based drugs and vaccines have been extensively studied to treat and prevent various diseases[3]. Thus, carbohydrate synthesis has attracted considerable attention owing to the numerous biological and pharmacological activities of carbohydrates. Glycosylation, the attachment of a glycosyl donor to a glycosyl acceptor via a glycosidic bond, is a central focus for synthesizing carbohydrates and is characterized by challenging reactions. The glycosyl donor bearing a leaving group at the anomeric position significantly influences the outcome of glycosylation reactions[4–6]. In the past decades, a series of effective glycosyl donors, such as glycosyl halides[7], glycosyl imidates[8], thioglycosides[9], glycosyl phosphites/phosphates[10], glycals[11], benzoyl/phenyl glycosides[12,13], and others[14–17], have been used in the chemical synthesis of glycans[18–21]. However, at present, no general glycosylation method for glycan synthesis has been reported. Novel and efficient glycosyl donors are highly demanded in constructing complex glycans.

Visible-light-mediated reactions have proven to be reliable and efficient for organic synthesis[22–26]. Photo-driven glycosylation has attracted significant attention in organic and carbohydrate chemistry[27–50]. Currently, visible-light-induced glycosylation reactions are achieved by either photoactivating a photosensitizer or using a stoichiometric activator first to generate an intermediate, which facilitates the departure of the leaving group, forming the glycosyl cation (Fig. 1A). Compared with traditional chemical glycosylation reactions, photo-glycosylation has historically been underexplored[27–50]. Thus, we envisaged that a direct light-activation of a photoactive-leaving group might greatly improve the photochemical glycosylation efficiency. However, to the best of our knowledge, visible-light-promoted glycosylation via a photoactive glycosyl donor has not been reported. Thus, we directed our efforts toward designing a photoactive-leaving group to fill this gap.

[1]State Key Laboratory of Natural and Biomimetic Drugs, School of Pharmaceutical Sciences, Peking University, Xue Yuan Road No. 38, Beijing 100191, China. [2]Ningbo Institute of Marine Medicine, Peking University, Ningbo 315010, China. ✉e-mail: decai@bjmu.edu.cn

**Fig. 1 | Background and this work. A** Known visible-light-induced glycosylation reactions. **B** Tropolone-involed photocyclization and acyl-transfer reaction. **C** Photosensitizer-free visible-light-promoted glycosylation enabled by 2-glycosyloxy tropone donor (*this work*).

## Results

### Reaction Optimization

We commenced our investigation by examining the glycosylation reaction of glycosyl donor **1a** with glycosyl acceptor **2a** (Table 1 and Supplementary Tables 1–4). When a mixture of **1a** (0.075 mmol) and **2a** (0.05 mmol) in 1,2-dichloroethane was stirred in the absence of light for 3 h, no desired product **3a** was detected (Table 1, entry 1). To our delight, we indeed isolated the desired product **3a** in 22% isolated yield when the reaction was exposed to light with a wavelength of 310–320 nm (Table 1, entry 2). We systematically investigated the impact of various irradiation wavelengths, including 365–370 nm, 380–390 nm, 415–420 nm, 430–435 nm, and 450–465 nm (Table 1, entries 3–7). We found light irradiation was essential for this reaction. Visible light was superior to ultraviolet light. Finally, blue LEDs (15 W) were chosen as this reaction's light source, and the reaction yield was 67% (Table 1, entry 7). A decrease in illumination intensity led to a reduction in the reaction yield (Table 1, entries 8–10). In addition, a modest excess of glycosyl donor **1a** was necessary to achieve a satisfactory isolated yield (82%, Table 1, entry 11). A catalytic NaOTf

could shorten the reaction time and improve the reaction yield (91%, Table 1, entry 12). The role of NaOTf might provide an anion and generate a highly reactive glycosyl triflate[57].

In the following experiments, a more challenging reaction between donor **1b** and acceptor **2b** was examined (Table 1, entries 13–25; Supplementary Table 5). A yield of 71% for the formation of disaccharide **3b** suggests that there is room for improvement in the reaction conditions (Table 1, entry 13). As a result, we checked other salt triflates and sodium salts with different anions, such as NaB(3,5-$(CF_3)_2$Ph)$_4$, NaSO$_2$CF$_3$, NaBF$_4$, NaNTf$_2$, TMSOTf, Bu$_4$NOTf, KOTf, and NaSO$_4$Me (Table 1, entries 14–21; Supplementary Table 5). The catalytic performance of sodium triflate and trimethylsilyl triflate was superior to other additives (Table 1, entries 13 and 21). Although the control experiment showed that the reaction could still be promoted by TMSOTf in the dark for 12 h, it is evident that the overall reaction efficiency remained low (44%, Table 1, entries 22–25). This indicates that light plays an important role in the reaction. Considering the glycosylation efficiency on the disarmed donor, we would choose TMSOTf as the additive with the loading of 10 mol% (Table 1, entry 21). Solvents other than dichloromethane reduced the yield of the desired product **3a** (Supplementary Table 3). Moreover, the molecular sieves suppressed this glycosylation reaction (Table S4, entry 10; Supplementary Table 5, entries 23–24).

### Evaluation of acceptor scope for photo-*O*-glycosylation

Alongside the optimized reaction conditions, we investigated the acceptor scope of the visible-light-promoted *O*-glycosylation reaction (Fig. 2). First, a perbenzoylated 2-galactosyloxy tropone (disarmed donor)[58] was chosen to examine the scope of the acceptors. As summarized in Fig. 2A, primary, secondary, and tertiary alcohols were suitable glycosyl acceptors under the standard reaction conditions, resulting in excellent yields of the desired glycosides **3c**–**3g**. The reaction of perbenzoylated 2-galactosyloxy tropone with various glycosyl acceptors bearing a free hydroxyl group at C-6, C-2, C-3, and C-4 positions generated the disaccharides **3h**–**3l** with yields ranging from 88% to 96%. The glycosyl acceptor with a *p*-methylphenylthio-leaving group for further glycosylation reaction was also an effective reactant in this protocol[9], generating the corresponding product **3m** with 98% yield. Low-reactive acceptors, such as phenols and carboxylic acids, were all amenable substrates in the visible-light-promoted glycosylation reaction. 4-Methoxy phenol, 4-hydroxycoumarin, and 6-methoxy-2-naphthol underwent smooth coupling with the disarmed donor to generate the phenolic glycosides **3n**–**3p** in 92% to 96% yields. No *C*-glycoside byproducts arising from *O* → *C*-glycoside rearrangement were detected in the crude mixtures[59]. In addition, both aromatic acid and alkyl acid were effectively reacted under the standard reaction conditions, generating **3q** and **3r** with excellent yields.

### Evaluation of donor scope for photo-*O*-glycosylation

Second, the scope of the donors was explored for *O*-glycosylation (Fig. 2B). The efficiency of the transformation was well illustrated by the reaction of the disarmed glycosyl donors with the low-reactive glycosyl acceptors[58], especially the glycosyl acceptors bearing a free hydroxyl group at C-4 positions. A series of perbenzoylated 2-glycosyloxy tropone donors, including glucosyl, mannosyl, arabinosyl, xylosyl, rhamnosyl, ribosyl, and amino-glucosyl donors, successfully underwent visible-light-promoted *O*-glycosylation, generating the corresponding disaccharides (**4a**–**4i**) with excellent yields. The peracetylated galactosyl donor and glucosyl donor were appropriate substrates for glycosylation, albeit in moderate to good yields of the products (**4j**–**4l**, 62–76%). Some acetylated acceptors were isolated as the byproducts, with about 10% yields arising from an acetyl transfer from the donors. This type of byproduct was formed with a yield up to 42% when a thioglycoside donor was used[60]. The

## The capacity of the chromophoric leaving group to absorb light at the irradiation wavelength (preferably visible light) is one of the fundamental prerequisites for a photoactive glycosyl donor. Tropolone and its derivatives are a class of non-benzenoid aromatic compounds known for their diverse biological activities[51]. Tropolone itself exhibits fluorophoric properties[52,53]. Extensive research has been conducted on the photocyclization and photocycloaddition of tropolone (Fig. 1B)[54,55]. Tropolonate salts are suitable for acyl-transfer catalysts under visible-light conditions[56]. However, tropolone has rarely been employed in carbohydrate chemistry. We envisaged that tropolone might be an ideal photoactive-leaving group for visible-light-induced glycosylation.

In this work, we report an efficient visible-light-promoted glycosylation using 2-glycosyloxy tropones as the donors.

**Table 1 | Screening and optimization of the reaction conditions[a]**

| Entry | Conditions | Yield (%)[a] |
|---|---|---|
| 1 | No light, 3 h | **3a** (0) |
| 2 | 310-320 nm, 3 h | **3a** (22) |
| 3 | 365-370 nm, 3 h | **3a** (20) |
| 4 | 385-390 nm, 3 h | **3a** (65) |
| 5 | 415-420 nm, 3 h | **3a** (66) |
| 6 | 430-435 nm, 3 h | **3a** (65) |
| 7 | blue LEDs, 3 h | **3a** (67) |
| 8 | blue LEDs (10 W), 3 h | **3a** (59) |
| 9 | blue LEDs (5 W), 3 h | **3a** (20) |
| 10 | blue LEDs (3 W), 3 h | **3a** (10) |
| 11[b] | blue LEDs, 5 h | **3a** (82) |
| 12[c] | blue LEDs, NaOTf, 20 min | **3a** (91) |
| 13[d] | NaOTf, 2.5 h | **3b** (71) |
| 14[d] | NaB(3,5- (CF$_3$)$_2$Ph)$_4$, 6 h | **3b** (63) |
| 15[d] | NaSO$_2$CF$_3$, 2.5 h | **3b** (60) |
| 16[d] | NaBF$_4$, 12 h | **3b** (30) |
| 17[d] | NaNTf$_2$, 8 h | **3b** (41) |
| 18[d] | NaSO$_4$Me, 2 h | **3b** (50) |
| 19[d] | Bu$_4$NOTf, 8 h | **3b** (42) |
| 20[d] | KOTf, 8 h | **3b** (54) |
| 21[d] | TMSOTf, 0.5 h | **3b** (92) |
| 22[d] | TMSOTf, in the dark, 12 h | **3b** (44) |
| 23[d] | NaOTf, in the dark, 12 h | **3b** (0) |
| 24[d] | no additive, 12 h | **3b** (13) |
| 25[d] | no additive, no light, 24 h | **3b** (0) |

[a] Reaction conditions: **1a** (0.075 mmol), **2a** (0.05 mmol), ClCH$_2$CH$_2$Cl (2.0 mL), blue LEDs (15 W, 450–465 nm), argon atmosphere, and isolated yields. [b] **1a** (0.1 mmol). [c] NaOTf (10 mol%) as the additive. [d] **1b** (0.075 mmol), **2b** (0.05 mmol), ClCH$_2$CH$_2$Cl (2.0 mL), blue LEDs (15 W, 450–465 nm), additive (10 mol%), argon atmosphere, and isolated yields. Ratios of α/β were determined using ¹H NMR, α/β = 1.1/1~3.4/1.

glycosylation of perbenzylated 2-galactosyloxy tropone donor generated disaccharides **4m** with a mixture of α/β anomers with good yield. Participating solvents could facilitate the preferential formation of one anomer[6], that is, acetonitrile favors the β-anomer, while diethyl ether favors the α-anomer. *Tert*-Butyldiphenylsilyl (**4n**) was tolerated well under the reaction conditions. Notably, perbenzoylated 2-lactosyloxy tropone was a viable donor, yielding 78% of the desired trisaccharide **4o**.

**Synthesis of glycosyl phosphosaccharides**

Given the above results, we examined the impact of visible-light-promoted *O*-glycosylation on the preparation of glycosyl phosphosaccharides (Fig. 3A). Phosphoglycans represent an important class of glycopolymers on the outer membrane of bacteria, yeasts, and protozoa containing anomeric phosphodiester linkages[61]. The synthesis of glycosyl phosphosaccharides is intrinsically challenging because the glycosylation of a glycosyl donor with a phosphoric acid usually forms a pair of diastereoisomers, especially the construction of α-glycosyl phosphosaccharides[62,63]. We first explored the reactivity of perbenzoylated 2-glycosyloxy tropone donor with phosphate

nucleophiles. A series of β-glycosyl phosphosaccharides (**5a–5e**) were efficiently synthesized with moderate to good yields through glycosylation reaction of various phosphate acceptors, such as dibenzyl phosphate, 6-*O*-benzyloxyphosphoryl glucoside, 3-*O*-benzyloxyphosphoryl glucoside, serinyl phosphate and 5′-uridylic acid. To our delight, the exposure of perbenzylated 2-glucosyloxy tropone donor and diphenyl phosphate to visible-light irradiation in the absence of the additives led to a 91% yield of glycosyl phosphate **5f** with high α-stereoselectivity (α/β > 20/1). Different glycosyl donors (galactosyl, mannosyl, xylosyl, and 2-azido-2-deoxy-glucosyl donors) effectively reacted under visible-light irradiation, generating the α-glycosyl phosphate **5g–5k** with good yields (57–85%). Notably, good stereoselectivity was achieved for the formation of α-diglycosyl phosphate **5l** (phosphoric acid as the acceptor). The above reaction was also effective for pyranose-6/2/4-*O*-phosphate to generate α-glycosyl phosphosaccharide **5m–5p** with high anomeric selectivity and good yields by changing different additives. Furthermore, serine-*O*-phosphate and 5′-uridylic acid were suitable glycosyl acceptors to generate the corresponding α-glycosyl phosphates **5q–5r** with high anomeric selectivity.

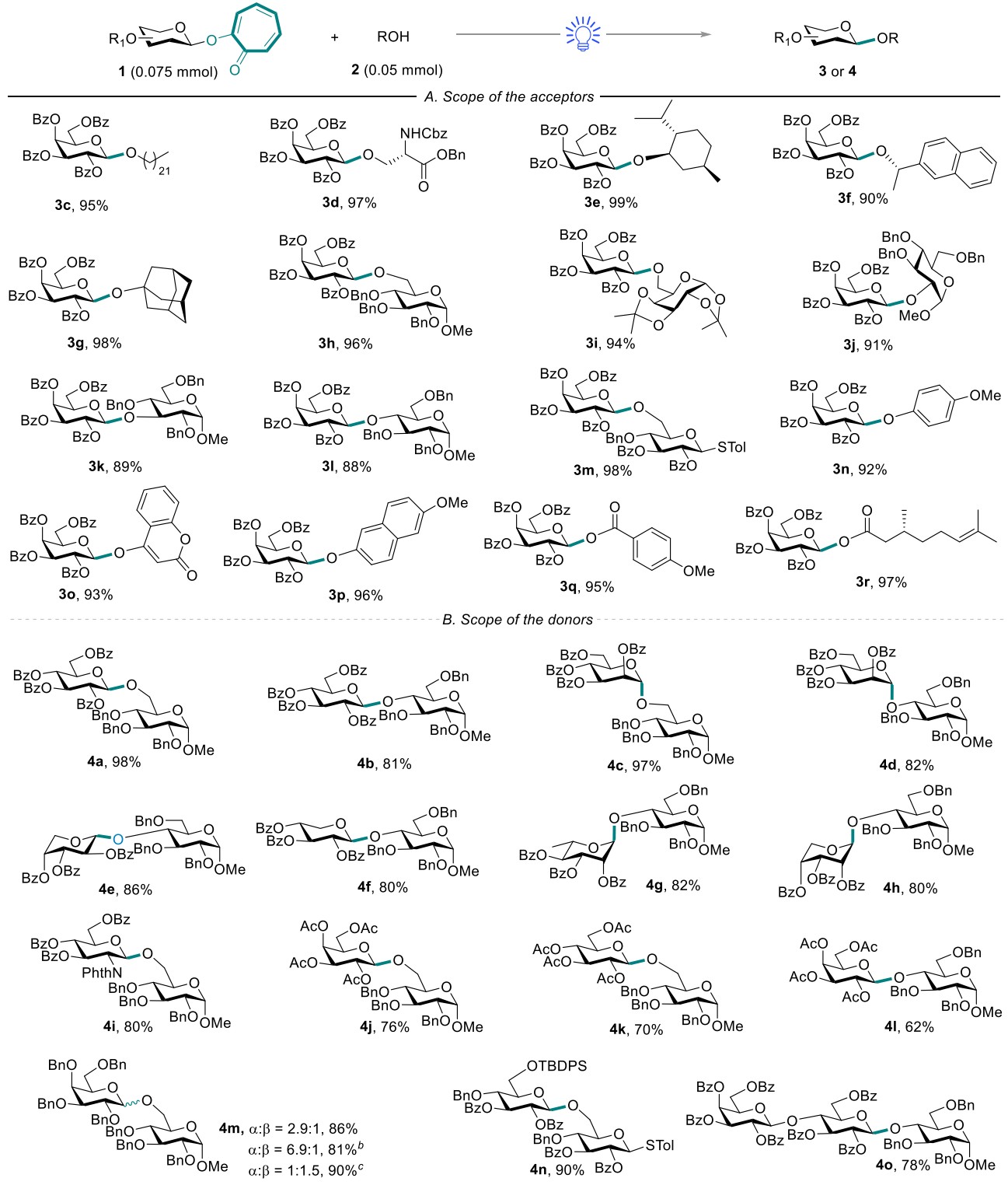

**Fig. 2 | Scope of the substrates for *O*-glycosylation[a]. A** Reaction conditions unless otherwise specified: TMSOTf (10 mol%), ClCH₂CH₂Cl (2.0 mL), blue LEDs (15 W), argon atmosphere, 0.5 h. **B** ClCH₂CH₂Cl: Et₂O = 1:5 (0.3 mL: 1.5 mL). [c] ClCH₂CH₂Cl: CH₃CN = 1:5 (0.3 mL: 1.5 mL).

## The substrate scope of photo-*N*-glycosylation

Our synthetic method was further demonstrated through *N*-glycosylation (Fig. 3B), which is another important aspect of evaluating the effectiveness of a new glycosyl donor[64–66]. After the pre-trimethylsilylation of pyrimidines with bis(trimethylsilyl)tri-fluoroacetamide (BSTFA) to improve the solubility of pyrimidines, we observed that pyranosyl-/furanosyl-donors, including D-galacto-, D-

gluco-, D-arabino-, and D-ribo-, effectively reacted with pyrimidines to form the corresponding *N*-glycosides **6a**–**6d** with excellent yields (94-98%). Moreover, the *N*-glycosylation effectively proceeded with pur-ines, such as 6-chloro-2-fluoropurine, 2,6-dichloropurine, and 6-chloropurine, resulting in satisfactory yields (90–95%) of N-9 nucleo-side **6e**–**6h**. All these results showed that 2-glycosyloxy tropones are suitable donors for *N*-glycosylation.

**Fig. 3 | Substrate scope for the stereoselective preparation of glycosyl phosphosaccharides and *N*-glycosylation[a]. A** Stereoselective synthesis of the glycosyl phosphates and glycosyl phosphosaccharides. **B** Synthesis of *N*-glycosides. **a** Reaction conditions unless otherwise specified: NaOTf (10–50 mol%), ClCH₂CH₂Cl (2.0 mL), blue LEDs (15 W), argon atmosphere, 2–12 h. **b** ClCH₂CH₂Cl (2.0 mL), blue LEDs (15 W), argon atmosphere, 0.5–4 h. Yields of α/β mixtures (α/β

mixtures were not separable), and ratios were determined by $^{31}$P NMR. **c** NaSO₄Me (30 mol%), blue LEDs (15 W), 1.5–8 h. **d** blue LEDs (15 W), 100 °C, 10–14 h. **e** Bis(trimethylsilyl)trifluoroacetamide (BSTFA, 4.0 equiv), ClCH₂CH₂Cl: CH₃CN = 1:1 (1.0 mL: 1.0 mL), argon atmosphere, 0.5 h; then TMSOTf (10 mol%), blue LEDs (15 W), 0.5–2 h, isolated yield.

**A. Gram-scale synthesis and modification.**

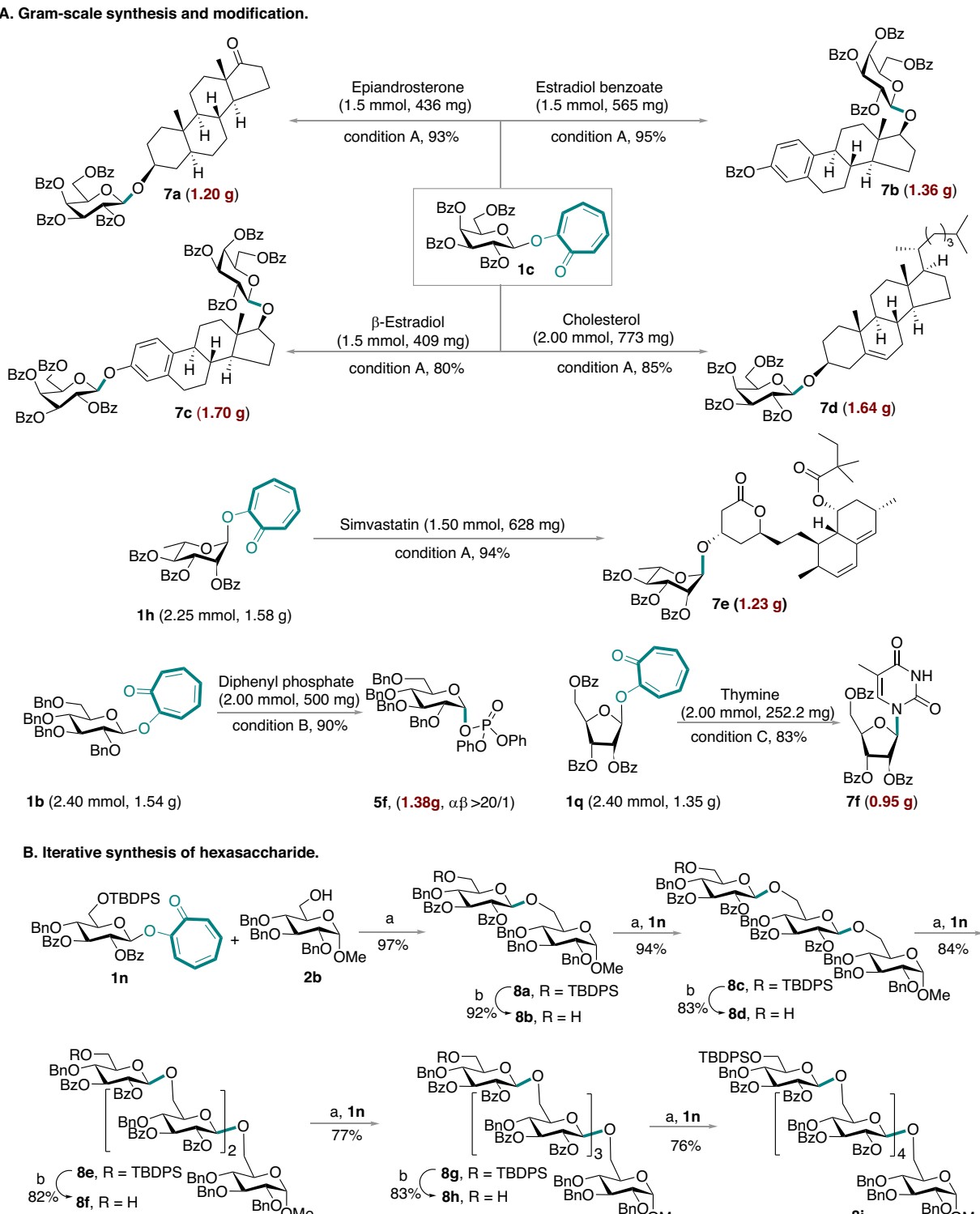

**Fig. 4 | Late-stage glycosylation and gram-scale synthesis. A** Gram-scale synthesis and modification. Conditions A: TMSOTf (10 mol%), ClCH₂CH₂Cl, blue LEDs (15 W), 2–24 h, argon atmosphere, isolated yield. Conditions B: ClCH₂CH₂Cl, blue LEDs (15 W), 2 h, argon atmosphere; Yields of α/β mixtures (α/β mixtures were not separable), and ratios were determined by $^{31}$P NMR. Conditions **C** TMSOTf (10 mol%), BSTFA (4.0 equiv), ClCH₂CH₂Cl: CH₃CN = 1:1, blue LEDs (15 W), 1 h, argon atmosphere, isolated yield. **B** Iterative synthesis of hexasaccharide. Reaction conditions: (**a**) TMSOTf (10 mol%), ClCH₂CH₂Cl, blue LEDs (15 W), argon atmosphere; **b** TBAF in THF (1 mol/L, 1.5 equiv), THF (2.0 mL).

## Late-stage glycosylation and gram-scale synthesis

As late-stage glycosylation is desirable in modern drug discovery[3,67], glycosylation of several natural products and pharmaceuticals was evaluated on a gram-scale (Fig. 4A). The mono/ di-glycosylation of perbenzoylated 2-galactosyloxy tropone (**1c**) with epiandrosterone, estradiol benzoate, β-estradiol, and cholesterol generated the corresponding glycosides **7a**–**7d** (1.20–1.70 g) with 80–95% yields. Moreover, glycosylation of

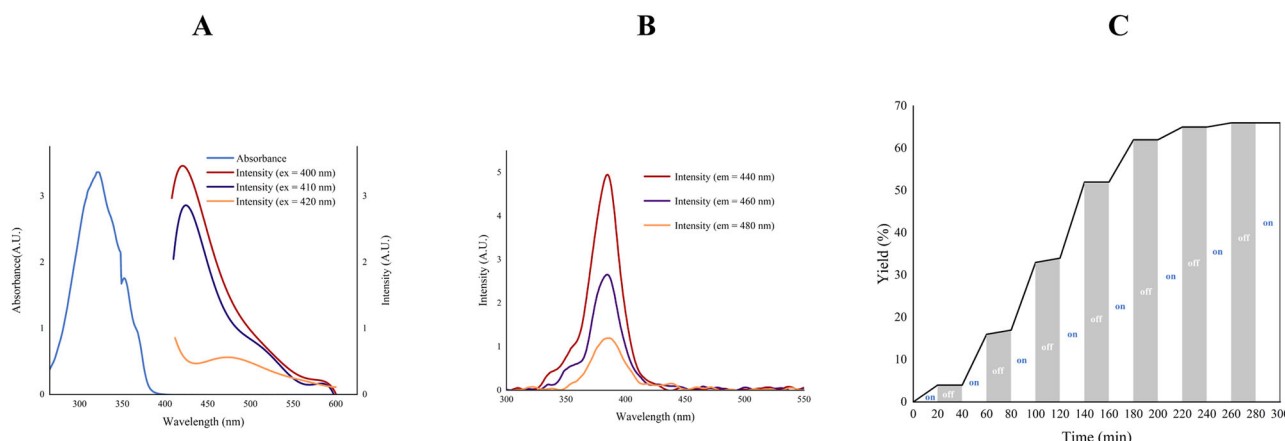

**Fig. 5 | Mechanistic investigations. A** UV−vis absorption and photoluminescence spectra of **1a**.[a] **B** The photoluminescence excitation spectra of **1a**.[a] **C** Light on-and-off experiments using the glycosylation of **1a** with **2a**. [b] [a] 0.10 mg/ mL of **1a** in ClCH$_2$CH$_2$Cl. [b] **1a** (1.5 equiv.), **2a** (1.0 equiv.), 1,3,5-trimethoxybenzene, ClCH$_2$CH$_2$Cl, blue LEDs (15 W), yields were determined by [1]H NMR.

perbenzoylated 2-L-rhamnosyloxy tropone (**1h**) with Simvastatin yielded 94% of gram-scale glycoside **7e**. Furthermore, α-glycosyl phosphate **5f** and nucleoside **7f** were obtained via the visible-light-promoted glycosylation on a gram-scale with excellent stereoselectivity and good yields. These successful applications strongly demonstrate the high practicability and the ease of scalability of our methodology.

Interesting, 2-glucosyloxy tropone can indeed serve as a glycosyl acceptor in the presence of molecular sieves, making it a valuable tool for orthogonal glycosylation reactions. 2-glucosyloxy tropone bearing a free hydroxyl group at C-6 (S27 in the Supplementary Information) could undergo glycosylation reactions with thioglycoside donor or glycosyl imidate donor, resulting in the formation of disaccharide products (S29 and S31 in the Supplementary Information) with moderate to good yields.

### Hexasaccharide synthesis

The ultimate aim of a glycosylation method is to synthesize glycans[4–6,18–21]. Thus, a hexasaccharide **8i** was prepared via iteratively visible-light-promoted glycosylation (Fig. 4B). *O*-glycosylation of 2-glucosyloxy tropone **1n** with the acceptor **2b** conducted under the standard condition yielded 97% of the disaccharide **8a**. The deprotection of the TBDPS group used stoichiometric amounts of tetrabutylammonium fluoride to generate the acceptor **8b**. The desired hexasaccharide **8i** was finally obtained via another four photo-glycosylation/desilylation cycles in a satisfactory isolated yield.

### Mechanistic studies

To gain more insight into visible-light-promoted glycosylation, we conducted a series of mechanistic experiments. The model reaction and control experiments are presented in Fig. 5 and Table 1. The experiment results revealed the significant impact of visible light irradiation on the glycosylation reaction. The glycosylation reaction time was significantly reduced when triflate was used to catalyze the reaction. UV−Vis absorption and photoluminescence spectra of **1a** revealed that the donor can be excited effectively by absorbing visible light (Fig. 5A, B, Supplementary Figs. 1–2)[52,53]. The light on-and-off experiments revealed that the glycosylation reaction occurred only under photo-irradiation, indicating that light played an important role in this reaction (Fig. 5C, Supplementary Table 7, Supplementary Figs. 4−5). Tropolone, originating from the leaving group departure, could be isolated as the byproduct[68,69]. The stereochemical outcome of the glycosyl phosphosaccharides might be attributed to the

continuous generation of a highly reactive and low concentration of β-glycosyl triflate[70]. Consequently, a mechanism involving direct excitation of the donor, aromatic resonance, departure of the leaving group to produce a glycosyl cation, and glycosylation was proposed (Supplementary Fig. 7).

## Discussion

In summary, we developed a versatile visible-light-driven glycosylation method using the 2-glycosyloxy tropone donor. This method was highly effective under the direct visible-light activation of the tropolone-leaving group at ambient temperature without a photosensitizer or activator. In addition, the photo-mediated protocol proved effective in enabling both the *O*-glycosylation and the *N*-glycosylation under mild conditions. The glycosylation method can accommodate a wide range of functionally complex acceptors with good scopes, providing a practical approach to glycoside synthesis. Particularly, the reaction demonstrated high versatility with a wide range of phosphates, thus generating glycosyl phosphates/phosphosaccharides with excellent anomeric selectivity. This method is not only suitable for the late-stage glycosylation of natural products and pharmaceuticals on gram scales but also for the iterative synthesis of hexasaccharide. The efficiency, robustness, and selectivity of the presented method make it an attractive protocol for synthesizing glycans and glycoconjugates.

## Methods

### General procedure for *O*-glycosylation

The glycosyl donor **1** (0.075 mmol), acceptor **2** (0.05 mmol) and additive (0.005 mmol) were dissolved in dry ClCH$_2$CH$_2$Cl (2.0 mL). The mixture was irradiated by blue LEDs at ambient temperature for 0.5 h. Upon completion, the solvent was concentrated under reduced pressure. The resulting residue was eluted by flash column chromatography (petroleum ether/EtOAc) to afford the glycosylated product **3, 4, 7** and **8**.

### General Procedure for glycosyl phosphosaccharides

The glycosyl donor **1** (0.06 mmol), acceptor **2** (0.05 mmol) and additive were dissolved in dry ClCH$_2$CH$_2$Cl (2.0 mL). The mixture was irradiated by blue LEDs at ambient temperature. Upon completion, Et$_3$N was added to quench the reaction and the solvent was concentrated under reduced pressure. The resulting residue was eluted by flash column chromatography (petroleum ether/EtOAc) to afford the glycosylated product **5**.

## General Procedure for *N*-glycosylation

To a solution of purines or pyrimidines (0.05 mmol) in anhydrous CH$_3$CN (1.0 mL) was added BSTFA (0.20 mmol) under Ar atmosphere. The suspension was stirred at room temperature until it becomes a clear solution. Then the **1** (0.06 mmol) dissolved in ClCH$_2$CH$_2$Cl (1.0 mL) and additive (0.005 mmol) were added to the above solution through a syringe, and the mixture was irradiated by blue LEDs at ambient temperature. Upon completion, the reaction mixture was concentrated in vacuo. The residue was purified by flash column chromatography on silica gel (petroleum ether/EtOAc) to obtain **6**.

## Data availability

The data supporting the findings of this study are available described in the article and its Supplementary Information Files are available from the corresponding authors on request.

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

## Acknowledgements

This work was financially supported by the National Natural Science Foundation of China (No. 21977003), the National Key R&D Program of China (No. 2022YFC3400800), and the Fundamental Research Funds for the Central Universities. We are grateful to the following Professors: Houhua Li, Xinjing Tang, Song Song, and Suwei Dong, for their insightful discussions.

## Author contributions

D.-C.X. conceived the research. J.Z. performed the main experiments. Z-X.L., X.W., C-F.G., P-Y.W., J-Z.C., and M.L., M.L. synthesized some building blocks. D.-C.X. and J.Z. analyzed the data. D.-C.X., J.Z., and X.-S.Y. wrote the manuscript.

## Competing interests

The authors declare no competing interest.
