## [Peer Review File · Nature Communications]

REVIEWER COMMENTS

Reviewer #1 (Remarks to the Author):

The Authors report on a novel approach to chemical O- and N-glycosylation using 2-glycosyloxytropones as donors. Visible-light irradiation may result in chemically reactive excited states that react to eventually form glycosidic products. The work consists of an optimization table in addition to substrate scope studies (57 entries including O- and N-glycosylation), gram-scale syntheses, an iterative hexasaccharide synthesis, and some mechanistic work.

In reading through this work, my level of interest was initially high. This is a highly novel concept, and the idea of a glycosylation donor that is activated purely with visible light and without any additives is VERY appealing. I can imagine this being a very useful method e.g. using the automated photochemical system recently introduced by this group. This is also a mechanistically novel transformation. However, as I continued with my review, a number of questions arose in my mind. In particular, there are some very critical controls missing from this study, and the extensive use of TMSOTf as an additive served to lessen my enthusiasm.

At the moment, I am not sure what my recommendation is. Therefore, I will be willing to reconsider this work if the Authors can provide meaningful responses to the following concerns.

manuscript:

1. First and foremost, I am not aware of any controls in which the 2-glycosyloxytropone donors and acceptors are combined with the additives NaOTf and TMSOTf IN THE DARK. There is a substantial rate increase with the use of both of these additives, and this immediately raises the question as to whether these are truly photochemical processes. While my sense of things is that NaOTf won't work in the dark, I think that it is a distinct possibility that TMSOTf WILL. Silylation or protonation of the carbonyl oxygen results in an intermediate with a substantial degree of tropylium (i.e. aromatic) character. Such a readily formed intermediate may facilitate glycosylation.
2. It is also rather disquieting to me that the Authors make no mention of the use of TMSOTf in their optimization chart in the manuscript itself but then quietly switch to its use in the substrate scope study. As the Authors know quite well, glycosyl trichloroacetimidates and N-phenyltrifluoroacetimidates are activated using catalytic amounts of TMSOTf without the need for visible-light irradiation. If the best results can only be obtained using TMSOTf as an additive, then what is the point of using glycosyloxytropones instead of the aforementioned imidate donors which are synthesized more easily (1 step from lactol) than glycosyloxytropones (at least 2 and as many as 3 steps from lactol)? This is a fact that I might be able to abide if this work were being submitted to Organic Letters or the Journal of Organic Chemistry, but the bar is very high for a publication like Nature Communications. The Authors will need to make a VERY good case for using TMSOTf or perhaps repeat their studies with NaOTf or with no additive.
3. A 57-entry substrate scope study is an exceedingly high number for a journal that specializes in Communications. The volume of work here reflects a Full Paper, and thus this might be more properly suited for an Article in JACS or a Research Article in Angewandte Chemie. However, I leave it to the discretion of the Editor to decide on the merits of this particular criticism.
4. In the Abstract, the Authors assert that this work is both "green" and "mild." I encourage the Authors to reconsider whether this work is truly green especially since they are using lots of

chlorinated solvent and are using a rather energy-intensive and non-solar source of light. Likewise, while "mild" is a highly subjective term, there are few organic chemists that would call TMSOTf a "mild" reagent.

5. In Figure 5a, the Authors indicate absorption and photoluminescence spectra of 1a. However, the absorption spectrum is not shown. Please include it! Further, it would be helpful if the Authors state here that the photoluminescence spectra are actually excitation spectra at the various indicated wavelengths. They are not photoluminescence spectra. There are no emission (fluorescence in this case?) spectra indicated in Figure 5, and those should also be included in Figure 5 as they are in Figure S1.

6. The Authors have missed a number of interesting observations in Figure 5b. There appears to be an acceleration in rate especially between 160 and 180 minutes and then again from 200 to 220 minutes. What is going on here? The rate seems to plateau off after this. At the bare minimum, the Authors should acknowledge these observations. Further, the Authors argue that the reaction only proceeds under irradiation. This is clearly not the case as the reaction is proceeding measurably between 140 and 160 minutes, 180 and 200 minutes, and 220 and 240 minutes. I think that the Authors are missing out on some observations that could prove to be very informative in understanding this transformation, however, there are other problems with the Authors' approach to this experiment. I will comment further on this experiment below.

7. On page 9, column 2, the Authors state that the "stereochemical outcome might be attributed to the continuous generation of a highly reactive and low concentration of β -glycosyl triflate 70." However, most of the examples in the substrate study are attributable to neighboring group participation from ester at the 2-position.

SI section:

1. The Authors should ensure that the number of significant figures in their measured quantities matches the number of significant figures in the mmoles reported. I've seen several errors of this nature in the SI section.

2. Under heading 5 (Glycosylation reactions using 2-glycosyloxy tropone donors), the Authors indicate in the SI that donor, acceptor, and NaOTf are "dissolved" in 1,2-dichloroethane. Is it actually true that NaOTf dissolves in 1,2-dichloroethane? I would not have expected this to be the case. Under this same heading, the Authors indicate that reactions are quenched with triethylamine. How much triethylamine was used?

3. In the light on/off experiment, the Authors indicate that acceptor 2a is used as an internal standard. This is a very unorthodox approach as the internal standard should be inert to the conditions. The Authors should repeat this experiment with an inert internal standard. Then, they should look for evidence of the reaction proceeding in the dark as appears to be the case in the first experiment.

4. In part 6.4 of the SI, the Authors indicate a "Plausible Mechanism." I wish that Authors in this community would stop using this term. "Mechanistic Hypothesis" or "Proposed Mechanism" are far better terms because they are less subjective. With regard to the mechanistic hypothesis, the Authors indicate that the excited state (top center) is in equilibrium (equilibrium arrows are indicated) with a ground state indicated with a set of zwitterionic resonance contributors (at right). Clearly, this cannot

be the case if irradiation is required to access the excited state. The zwitterionic resonance structures only serve to confuse, in the opinion of this Reviewer. Further, the Authors indicate NaOTf in this mechanistic proposal whereas they conduct all of their mechanistic work in its absence. In the opinion of the Reviewer, this mechanistic hypothesis should be removed from the SI. I don't think that the Authors know enough yet to warrant a mechanistic proposal.

Reviewer #2 (Remarks to the Author):

In this manuscript, Xiong and coworkers reported a novel tropolone derived glycosyl donor. The donor could absorb blue light and enable glycosylation without other activating reagents. Catalytic NaOTf was then discovered for promoting the yield significantly. Other control reactions and light on-off experiments have shown that the continuous irradiation is necessary for the completion of the reaction. The reaction scope is phenomenal. Thus, I recommend this manuscript publish in Nature Commun. after the authors addressed the following questions:

1. Since this is a new glycosyl handle, does it orthogonal with other glycosyl handles? For example, If treating this glycosyl donor with BF₃OEt or TfOH, does glycosylation happen?
2. I believe it would be better to include the proposed mechanism in the article (instead of SI).
3. This is a minor one. The photocyclization reaction of tropolone has a long history (ref. 54). It is encouraged that the authors cite some recent synthetic literatures, but the ref. 55 is not suitable. The ref. 55 is not discussing photocyclization/photocycloaddition of tropolone.

Response to the comments and suggestions

Reviewer #1:

The Authors report on a novel approach to chemical O- and N-glycosylation using 2-glycosyloxytropone as donors. Visible-light irradiation may result in chemically reactive excited states that react to eventually form glycosidic products. The work consists of an optimization table in addition to substrate scope studies (57 entries including *O*- and *N*-glycosylation), gram-scale syntheses, an iterative hexasaccharide synthesis, and some mechanistic work.

In reading through this work, my level of interest was initially high. This is a highly novel concept, and the idea of a glycosylation donor that is activated purely with visible light and without any additives is VERY appealing. I can imagine this being a very useful method e.g. using the automated photochemical system recently introduced by this group. This is also a mechanistically novel transformation. However, as I continued with my review, a number of questions arose in my mind. In particular, there are some very critical controls missing from this study, and the extensive use of TMSOTf as an additive served to lessen my enthusiasm.

At the moment, I am not sure what my recommendation is. Therefore, I will be willing to reconsider this work if the Authors can provide meaningful responses to the following concerns.

We would like to express our sincere appreciation for the careful review of our manuscript and the insightful questions and suggestions raised. We highly value your concerns and have taken them into careful consideration. In the following section, we provide a point-by-point response to each of your comments. We hope that our responses adequately address your concerns.

Reviewer 1, comment 1: First and foremost, I am not aware of any controls in which the 2-glycosyloxytropone donors and acceptors are combined with the additives NaOTf and TMSOTf IN THE DARK. There is a substantial rate increase with the use of both of these additives, and this immediately raises the question as to whether these are truly photochemical processes. While my sense of things is that NaOTf won't work in the dark, I think that it is a distinct possibility that TMSOTf WILL. Silylation or protonation of the carbonyl oxygen results in an intermediate with a substantial degree of tropylium (i.e. aromatic) character. Such a readily formed intermediate may facilitate glycosylation.

Response: We appreciate your valuable suggestion, as it raises an important point.

The results of the control experiments have been incorporated into Table 1 in the manuscript and Supplementary Table S5. We investigated the glycosylation reaction of perbenzylated 2-glucosyloxy tropone **1b** and glycosyl acceptor **2b** using the additives NaOTf or TMSOTf under dark conditions. When NaOTf was used as a catalyst for the reaction between **1b** and **2b** to obtain the desired product **3b**, a yield of 71% was achieved (Table 1, entry 14). However, no product was detected when NaOTf was used in the absence of light (Table 1, entry 24). On the other hand, the desired product **3b** was obtained with a yield of 92% when TMSOTf was used as the additive (Table 1, entry 22). By contrast, when TMSOTf was used in the dark, the yield of **3b** was just 44%, and this yield remained unchanged even with an extended reaction time (Table 1, entry 23).

The photochemical nature of the process is evident from three aspects. Firstly, glycosylation can occur solely under light conditions, as demonstrated by Table 1, entries 1-12 and 25. Secondly, the use of TMSOTf without light leads to a low yield, as shown in Table 1, entry 23. Lastly, decomposition experiments of the donor in the presence of light further support the photoactivation process, as illustrated in Supplementary Fig. S3.

Table 1. Screening and optimization of the reaction conditions.^[a]

The reaction scheme shows the glycosylation of a tropone derivative (**1**) with a glycosyl acceptor (**2a** or **2b**) in the presence of a catalyst (**R**) and light, yielding products **3a** or **3b**. The reaction is performed in dichloroethane ($\text{ClCH}_2\text{CH}_2\text{Cl}$). The glycosyl acceptor **2a** is a perbenzylated glucose derivative, and **2b** is a perbenzylated glucose derivative with a methyl group at the C2 position. The catalyst **R** is either a perbenzylated glucose derivative (**1a**) or a perbenzylated glucose derivative with a methyl group at the C2 position (**1b**). The products **3a** and **3b** are the corresponding glycosylated tropone derivatives.

Entry	Conditions	Yield (%) ^[a]
1	No light, 3 h	3a (0)
2	310-320 nm, 3 h	3a (22)
3	365-370 nm, 3 h	3a (20)
4	385-390 nm, 3 h	3a (65)
5	415-420 nm, 3 h	3a (66)

6	430-435 nm, 3 h	3a (65)
7	blue LEDs, 3 h	3a (67)
8	blue LEDs (10 W), 3 h	3a (59)
9	blue LEDs (5 W), 3 h	3a (20)
11	blue LEDs (3 W), 3 h	3a (10)
12 ^[b]	blue LEDs, 5 h	3a (82)
13 ^[c]	blue LEDs, NaOTf, 20 min	3a (91)
14 ^[d]	NaOTf, 2.5 h	3b (71)
15 ^[d]	NaB(3,5-(CF ₃) ₂ Ph) ₄ , 6 h	3b (63)
16 ^[d]	NaSO ₂ CF ₃ , 2.5 h	3b (60)
17 ^[d]	NaBF ₄ , 12 h	3b (30)
18 ^[d]	NaN Tf ₂ , 8 h	3b (41)
19 ^[d]	NaSO ₄ Me, 2 h	3b (50)
20 ^[d]	Bu ₄ NOTf, 8 h	3b (42)
21 ^[d]	KOTf, 8 h	3b (54)
22 ^[d]	TMSOTf, 0.5 h	3b (92)
23 ^[d]	TMSOTf, in the dark, 12 h	3b (44)
24 ^[d]	NaOTf, in the dark, 12 h	3b (0)
25 ^[d]	no additive, 12 h	3b (13)
26 ^[d]	no additive, no light, 24 h	3b (0)

[a] Reaction conditions: **1a** (0.075 mmol), **2a** (0.05 mmol), ClCH₂CH₂Cl (2.0 mL), blue LEDs (15 W, 450-465 nm), argon atmosphere, and isolated yields. [b] **1a** (0.1 mmol). [c] NaOTf (10 mol%) as the additive. [d] **1b** (0.075 mmol), **2b** (0.05 mmol), ClCH₂CH₂Cl (2.0 mL), blue LEDs (15 W, 450-465 nm), additive (10 mol%), argon atmosphere, and isolated yields. Ratios of α/β were determined using ¹H NMR, $\alpha/\beta = 1.1/1 \sim 3.4/1$.

Table S5. Optimization of the glycosylation reaction between donor **1b** with acceptor **2b**.

Entry	Additive	Time (h)	Yield (%) ^[b]	α/β ratio ^[b]
1	no additive	12	13	1.7/1
2	NaOTf	2.5	71	1.8/1

3	NaB(3,5-(CF ₃) ₂ Ph) ₄	6	63	1.7/1
4	NaSO ₂ CF ₃	2.5	60	1.7/1
5	NaBF ₄	12	30	2.2/1
6	NaOAc	12	10	1.6/1
7	Sodium p -toluenesulfinate	12	23	1.4/1
8	NaI	12	18	1.7/1
9	NaClO ₄	12	10	1.7/1
10	NaN Tf ₂	8	41	1.7/1
11	NaOTs	12	10	1.7/1
12	NaSO ₄ Me	2	50	3.0/1
13	NaSO ₃ Me	10	10	1.7/1
14	Na ₂ SO ₄	12	10	1.7/1
15	Bu ₄ NOTf	8	42	2.0/1
16	KOTf	8	54	1.9/1
17	TfOH	0.5	84	1.9/1
18	BF ₃ ·Et ₂ O	0.5	17	3.4/1
19	HOPO(OPh) ₂	12	10	1.7/1
20	H ₃ PO ₄	1	17	1.1/1
21	TMSOTf	0.5	92	2.0/1
22	no Blue LEDs, no additive	24	NR	-
23	NaOTf, in dark	12	NR	-
24	TMSOTf, in dark	0.5	44	2.0/1
25	TMSOTf, in dark	12	44	2.0/1
26	TMSOTf, 4Å MS (200 mg)	12	13	2.0/1
27	TMSOTf, 4Å MS (200 mg), in dark	12	NR	-

[a] Reaction conditions: **1b** (0.075 mmol), **2b** (0.05 mmol), additive (10 mol%), blue LEDs (15 W), ClCH₂CH₂Cl (2.0 mL), argon atmosphere. [b] Isolated yields, ratios of α/β determined by ¹H NMR.

Supplementary Fig. S3 Decomposition experiments of the donor **1b** and **1c** in the absence/presence of light.

Reviewer 1, comment 2: It is also rather disquieting to me that the Authors make no mention of the use of TMSOTf in their optimization chart in the manuscript itself but then quietly switch to its use in the substrate scope study. As the Authors know quite well, glycosyl trichloroacetimidates and N-phenyltrifluoroacetimidates are activated using catalytic amounts of TMSOTf without the need for visible-light irradiation. If the best results can only be obtained using TMSOTf as an additive, then what is the point of using glycosyloxytropones instead of the aforementioned imidate donors which are synthesized more easily (1 step from lactol) than glycosyloxytropones (at least 2 and as many as 3 steps from lactol)? This is a fact that I might be able to abide if this work were being submitted to Organic Letters or the Journal of Organic Chemistry, but the bar is very high for a publication like Nature Communications. The Authors will need to make a VERY good case for using TMSOTf or perhaps repeat their studies with NaOTf or with no additive.

Response: We are grateful to the reviewer for bringing this matter to our attention and we sincerely apologize for the oversight in neglecting to include the screening of glycosylation conditions in the manuscript and Supplementary Information. We have rectified this issue by incorporating the comprehensive screening results into Table 1 in the manuscript and Supplementary Table S5.

The glycosylation conditions for combining glycosyl donor 1b with glycosyl acceptor 2b were included in Table 1 of the manuscript and Supplementary Table S5. The formation of disaccharide **3b** with a 13% isolated yield was observed when a mixture of **1b** (0.075 mmol) and **2b** (0.05 mmol) in 1,2-dichloroethane was subjected to blue LED irradiation under an argon atmosphere for 12 hours (Table 1, entry 25; Supplementary Table S5, entry 1). The catalytic reaction between **1b** and **2b** was facilitated by NaOTf, resulting in the desired product **3b** with a 71% yield (Table 1, entry 14; Supplementary Table S5, entry 2). Subsequently, the catalytic activities of sodium salts with various anions were examined (Table 1, entries 15-19; Supplementary Table S5, entries 3-14). It was observed that most sodium salts with different anions promoted the glycosylation reaction, with sodium triflate demonstrating superior catalytic performance compared to other sodium salts. Other triflate salts, such as Bu₄NOTf and KOTf, were found to be less effective in promoting glycosylation (Table 1, entries 22; Supplementary Table S5, entries 15-16). TMSOTf showed better effectiveness compared to NaOTf (Table 1, entries 20-21; Supplementary Table S5, entries 24-25). However, under dark conditions, TMSOTf lead to a low yield (Table 1, entry 23; Table S5, entry 21). Additives such as BF₃·Et₂O did not exhibit any significant effect on the glycosylation reaction (Supplementary Table S5, entries 18-20). Considering the glycosylation efficiency with the disarmed donor, TMSOTf was ultimately chosen as the additive (Supplementary Table S5, entry 21).

Glycosyloxypotropone offer several advantages. *Firstly*, 2-glycosyloxypotropone exhibits stability at room temperature, unlike glycosyl trichloroacetimidates. *Secondly*, 2-glycosyloxypotropone donors can be synthesized in a single step from lactol, as exemplified by the synthesis of compound **1b** described in the Supplementary Information and Scheme R1. *Thirdly*, 2-glycosyloxypotropone can be activated under various conditions, including NaOTf and TMSOTf (Tables 1 and Supplementary Table S5). We conducted studies on glycosyl trichloroacetimidates and *N*-phenyltrifluoroacetimidates donors using NaOTf as an additive, as well as without any

additive. Under classical conditions, the absence of an additive or replacement of TMSOTf with NaOTf resulted in no glycosylation reaction (Scheme R2). *Most importantly*, 2-glycosyloxytropone also functions as a glycosyl acceptor in the presence of molecular sieves, a capability that imitates lack (Scheme R3; S27, S29, and S31 in the Supplementary Information). The detailed results can be found on pages 10, and 72-75 of the Supplementary Information (SI). Scheme R1, R2 and R3 provide a summary of the findings.

Scheme R1. Synthesis of compound **1b**

Scheme R2. Activation of glycosyl trichloroacetimidates or *N*-phenyltrifluoroacetimidates donors using NaOTf.

Scheme R3. Glycosylation of 2-glycosyloxypyrone acceptor with thioglycoside or trichloroacetimidates.

Reviewer 1, comment 3: A 57-entry substrate scope study is an exceedingly high number for a journal that specializes in Communications. The volume of work here reflects a Full Paper, and thus this might be more properly suited for an Article in JACS or a Research Article in Angewandte Chemie. However, I leave it to the discretion of the Editor to decide on the merits of this particular criticism.

Response: We are grateful for the reviewer's comment to our work. We want to emphasize that Nature Communications is our top preference, and we have solely submitted this manuscript to Nature Communications. We have thoroughly reviewed the requirements outlined in the "brief guide to manuscript submission" on the official website of Nature Communications, and we did not come across any specific restrictions pertaining to substrate scope studies.

Reviewer 1, comment 4: In the Abstract, the Authors assert that this work is both "green" and "mild." I encourage the Authors to reconsider whether this work is truly green especially since they are using lots of chlorinated solvent and are using a rather energy-intensive and non-solar source of light. Likewise, while "mild" is a highly subjective term, there are few organic chemists that would call TMSOTf a "mild" reagent.

Response: We appreciate the comments provided by the reviewer regarding our work. In response, we have made revisions to the manuscript. Specifically, we have removed the term "green" and modified the phrase "The protocol featured mild conditions" to "The protocol featured straightforward conditions."

Reviewer 1, comment 5: In Figure 5a, the Authors indicate absorption and photoluminescence

spectra of **1a**. However, the absorption spectrum is not shown. Please include it! Further, it would be helpful if the Authors state here that the photoluminescence spectra are actually excitation spectra at the various indicated wavelengths. They are not photoluminescence spectra. There are no emission (fluorescence in this case?) spectra indicated in Figure 5, and those should also be included in Figure 5 as they are in Figure S1.

Response: We are grateful for the reviewer's comment to our work.

In the manuscript, Figure 5A presents the absorption, photoluminescence, and photoluminescence excitation spectra of compound **1a**. The photoluminescence spectra of compound **1a** were measured at various excitation wavelengths (400 nm, 410 nm, and 420 nm), as depicted in Figure 5A(a). Additionally, the photoluminescence excitation spectra of compound **1a** were obtained at different emission wavelengths (440 nm, 460 nm, and 480 nm), as shown in Figure 5A(b).

Figure 5A. (a) UV-vis absorption and photoluminescence spectra of **1a**

Figure 5A. (b) The photoluminescence excitation spectra of **1a**

Reviewer 1, comment 6: The Authors have missed a number of interesting observations in Figure 5b. There appears to be an acceleration in rate especially between 160 and 180 minutes and then again from 200 to 220 minutes. What is going on here? The rate seems to plateau off after this. At the bare minimum, the Authors should acknowledge these observations. Further, the Authors argue that the reaction only proceeds under irradiation. This is clearly not the case as the reaction is proceeding measurably between 140 and 160 minutes, 180 and 200 minutes, and 220 and 240 minutes. I think that the Authors are missing out on some observations that could prove to be very informative in understanding this transformation, however, there are other problems with the Authors' approach to this experiment. I will comment further on this experiment below.

Response: We sincerely appreciate the professional advice from the reviewer.

Taking into account the reviewer's suggestions, we conducted light on/off experiments using 1,3,5-trimethoxybenzene as the internal standard, as shown in Figure 5B. By making adjustments to the experimental protocol, we were able to rectify certain errors. Indeed, it was noted that the reaction rate did not exhibit any acceleration.

The observed reactions during the time intervals of 140 to 160 minutes, 180 to 200 minutes, and 220 to 240 minutes may be attributed to prior light exposure before conducting the NMR experiments. Nevertheless, by implementing rigorous light avoidance measures after sampling, we did not observe any subsequent reactions occurring.

Figure 5B. Light on and light off experiments

Reviewer 1, comment 7: On page 9, column 2, the Authors state that the "stereochemical outcome might be attributed to the continuous generation of a highly reactive and low concentration of β -glycosyl triflate⁷⁰." However, most of the examples in the substrate study are attributable to neighboring group participation from ester at the 2-position.

Response: We are grateful for the reviewer's comment on our work. The "stereochemical outcome" refers to the selective synthesis of glycosyl phosphates/phosphosaccharides. The benzyl-protected 2-glycosyloxytropone donors could be selectively used to synthesize α - glycosyl phosphates/phosphosaccharides. We have revised the "The stereochemical outcome might be attributed to the continuous generation of a highly reactive and low concentration of β -glycosyl triflate" to "The stereochemical outcome of the glycosyl phosphosaccharides might be attributed to the continuous generation of a highly reactive and low concentration of β -glycosyl triflate" in the manuscript.

Reviewer 1, comment 8: The Authors should ensure that the number of significant figures in their measured quantities matches the number of significant figures in the mmoles reported. I've seen several errors of this nature in the SI section.

Response: We appreciate the diligent review by the reviewer, and we acknowledge that the errors

brought to our attention were indeed oversights on our part. Upon a thorough review of the Supporting Information, we have identified and corrected the errors accordingly.

Reviewer 1, comment 9: Under heading 5 (Glycosylation reactions using 2-glycosyloxy tropone donors), the Authors indicate in the SI that donor, acceptor, and NaOTf are "dissolved" in 1,2-dichloroethane. Is it actually true that NaOTf dissolves in 1,2-dichloroethane? I would not have expected this to be the case. Under this same heading, the Authors indicate that reactions are quenched with triethylamine. How much triethylamine was used?

Response: We would like to express our gratitude to the reviewer for bringing this to our attention. We acknowledge that it was an oversight on our part, as we did not accurately describe this particular operation. In response, we have revised the description in the Supplementary Information to provide a more accurate and detailed account.

We would like to present an example of the revisions we have made: "The mixture of the glycosyl donor **1d** (42.0 mg, 0.06 mmol), acceptor **2q** (13.9 mg, 0.05 mmol) and NaOTf (0.9 mg, 0.005 mmol) in dry ClCH₂CH₂Cl (2.0 mL), was irradiated by blue LEDs at ambient temperature for 4 h."

In addition, Et₃N (0.1 mL) was added to the reaction.

Reviewer 1, comment 10: In the light on/off experiment, the Authors indicate that acceptor **2a** is used as an internal standard. This is a very unorthodox approach as the internal standard should be inert to the conditions. The Authors should repeat this experiment with an inert internal standard. Then, they should look for evidence of the reaction proceeding in the dark as appears to be the case in the first experiment.

Response: We appreciate the reviewer for bringing this matter to our attention. Following the reviewer's suggestions, we repeated the light on/off experiments using 1,3,5-trimethoxybenzene as the standard. The results of these experiments are presented in Figure 5B of the main text, as well as in Table S7, Figure S4, and Figure S5 of the Supplementary Information. Based on these new experimental findings, it has been observed that the reaction does not occur in the absence of light.

Table S7. Light on and light off experiments using the reaction of **1a** with **2a**.

Entry	Time (min)	Yield of 3a (%) ^a
1	20	4
2	40	4
3	60	16
4	80	16
5	100	33
6	120	33
7	140	52
8	160	52
9	180	62
10	200	62
11	220	65
12	240	65
13	260	66
14	280	66
15	300	66

[a] Yields were determined by ¹H NMR

Figure 5B. Light on and light off experiments

Reviewer 1, comment 11: In part 6.4 of the SI, the Authors indicate a "Plausible Mechanism." I wish that Authors in this community would stop using this term. "Mechanistic Hypothesis" or "Proposed Mechanism" are far better terms because they are less subjective. With regard to the mechanistic hypothesis, the Authors indicate that the excited state (top center) is in equilibrium (equilibrium arrows are indicated) with a ground state indicated with a set of zwitterionic resonance contributors (at right). Clearly, this cannot be the case if irradiation is required to access the excited state. The zwitterionic resonance structures only serve to confuse, in the opinion of this Reviewer. Further, the Authors indicate NaOTf in this mechanistic proposal whereas they conduct all of their mechanistic work in its absence. In the opinion of the Reviewer, this mechanistic hypothesis should be removed from the SI. I don't think that the Authors know enough yet to warrant a mechanistic proposal.

Response: We are grateful for the reviewer's comment on our work. Based on the current data, we believe that it is possible to propose a mechanism. In order to facilitate better understanding, we have included a proposed mechanism in Figure S6 of the Supplementary Information.

Reviewer #2:

In this manuscript, Xiong and coworkers reported a novel tropone derived glycosyl donor. The donor could absorb blue light and enable glycosylation without other activating reagents. Catalytic NaOTf was then discovered for promoting the yield significantly. Other control reactions and light on-off experiments have shown that the continuous irradiation is necessary for the completion of the reaction. The reaction scope is phenomenal. Thus, I recommend this manuscript publish in Nature Commun. after the authors addressed the following questions:

We are deeply grateful for the comments provided by the reviewer regarding our manuscript.

Reviewer 2, comment 1: Since this is a new glycosyl handle, does it orthogonal with other glycosyl handles? For example, If treating this glycosyl donor with $\text{BF}_3\cdot\text{OEt}_2$ or TfOH, does glycosylation happen?

Response: These are highly thought-provoking questions. To address these, we utilized 2-glycosyloxypone **S27** as the receptor. Our findings indicate that the presence of molecular sieves has a suppressive effect on our glycosylation reaction (Supplementary Table S4, Supplementary entry 10; Supplementary Table S5, entries 26-27). Thus, we successfully accomplished orthogonal glycosylation reactions involving thioglycosides or glycosyl trichloroacetimidates. Detailed results can be found on pages 72-75 of the Supplementary Information (SI), and are summarized in Scheme R4.

The following statement has been added to the main text: “Interesting, 2-glycosyloxy tropone can indeed serve as a glycosyl acceptor in the presence of molecular sieves, making it a valuable tool for orthogonal glycosylation reactions. 2-glycosyloxy tropone bearing a free hydroxyl group at C-6 (**S27** in Supplementary Information) could undergo glycosylation reactions with thioglycoside donor or glycosyl imidate donor, resulting in the formation of disaccharide products (**S29** and **S31** in Supplementary Information) with moderate to good yields.”

Scheme R4. 2-Glycosyloxypnone **S27** as the acceptor.

In addition, thioglycosides could act as an acceptor in our reaction system (**3m**, Figure 2 in the manuscript).

$\text{BF}_3 \cdot \text{OEt}_2$ or TfOH can serve as catalyst in our photoglycosylation. However, We treated glycosyl donor **1b** and glycosyl acceptor **2b** with $\text{BF}_3 \cdot \text{OEt}_2$ or TfOH, no glycosylation occurred in the presence of molecular sieves (Scheme R5).

Scheme R5. Glycosylation of 2-glycosyloxypnone donor using $\text{BF}_3 \cdot \text{OEt}_2$ or TfOH as the catalyst in the presence of molecular sieves.

Reviewer 2, comment 2: I believe it would be better to include the proposed mechanism in the

article (instead of SI).

Response: We greatly appreciate your suggestion. Considering the limitations of the main text, we have included the proposed mechanism in the Supplementary Information (SI).

Reviewer 2, comment 3: This is a minor one. The photocyclization reaction of tropolone has a long history (ref. 54). It is encouraged that the authors cite some recent synthetic literatures, but the ref. 55 is not suitable. The ref. 55 is not discussing photocyclization/photocycloaddition of tropolone.

Response: We appreciate the valuable suggestion provided by the reviewer. As a result, we have replaced the previous reference 54 and 55.

“54. Coote, S. C. 4- π -Photocyclization: scope and synthetic applications. *Eur. J. Org. Chem.* 1405-1423 (2020).

55. Ge, Z.-P. et al. Cephalodiones A–D: compound characterization and semisynthesis by [6+6] cycloaddition. *Angew. Chem. Int. Ed.* **60**, 9374-9378 (2021).

REVIEWER COMMENTS

Reviewer #1 (Remarks to the Author):

I am mostly satisfied with the Authors' rebuttal. I appreciate their efforts to clarify the introduction of TMSOTf into their studies, and I agree with them that visible-light irradiation plays an important role in the process. The new proposed mechanism is more conservative and may be justified by the Authors' observations. Please see the last item in this Review to address mechanistic considerations.

I will be in favor of publication if the Authors can address the following concerns in a satisfactory way:

1. In the Abstract, the Authors call this process "activator-free." Considering the rate/yield enhancements engendered with the addition of NaOTf and TMSOTf, this is not an accurate statement. Please remove it.

2. On page 10, column 1, toward the bottom, the Authors assert that the failure of rxn to progress in the dark demonstrates that this is not a chain mechanism. Yoon and co-worker have shown, conclusively, that this kind of claim is erroneous since some chain processes terminate very quickly without irradiation. Please remove this claim. Here is Yoon's work which the Authors should read: <https://pubs.rsc.org/en/content/articlelanding/2015/sc/c5sc02185e>

Neither Yoon nor Cismesia is the Reviewer.

3. Most importantly, in the SI, the Authors have failed to match the number of significant figures in their measured quantities to the number of significant figures in the number of millimoles reported. In the preparation of 1c, for instance, the Authors indicate that 2.50 g (3 significant figures) of S2 correlates to 20.00 millimoles (4 significant figures). The number of significant figures should match. In the preparation of 1d, the Authors indicate that 0.81 g (2 significant figures) of S4 correlates to 5.09 millimoles (3 significant figures) of compound. The number of significant figures should match. At a glance, I can see dozens of mistakes like this throughout the SI. Please correct them. I will hold up the publication of this work for as long as is necessary until this problem is corrected.

4. Finally, I have noticed something that could be interesting in Figure 5. Note that the slope of the line going from 0-20 minutes is actually smaller than 40-60 minutes. It would appear that there is an induction period that one would not expect if the mechanistic hypothesis in the SI is correct. Is it possible, for instance, that free 2-hydroxytropolone forms an exciplex with the substrate donor and that this is more reactive than substrate donor alone? The Authors could consider adding 2-hydroxytropolone to their reaction mixtures to see if this enhances rate. While such an experiment is not something that I deem necessary for the current publication, it might enrich the work if the answer to my question is "yes."

Reviewer #2 (Remarks to the Author):

In this revised manuscript, authors not only answered all my questions, but also include more useful information along with addressing reviewers' comments. Thus, I recommend the manuscript publish in Nature Communications.

Response to the comments and suggestions.

Reviewer #1:

I am mostly satisfied with the Authors' rebuttal. I appreciate their efforts to clarify the introduction of TMSOTf into their studies, and I agree with them that visible-light irradiation plays an important role in the process. The new proposed mechanism is more conservative and may be justified by the Authors' observations. Please see the last item in this Review to address mechanistic considerations. I will be in favor of publication if the Authors can address the following concerns in a satisfactory way:

We would like to extend our sincere appreciation for the thorough review of our manuscript and the insightful questions and suggestions that were raised. In the upcoming section, we aim to provide a comprehensive response to each of your comments, addressing them point by point. We hope that our responses sufficiently address the concerns you have expressed.

Reviewer 1, comment 1: In the Abstract, the Authors call this process "activator-free." Considering the rate/yield enhancements engendered with the addition of NaOTf and TMSOTf, this is not an accurate statement. Please remove it.

Response: We greatly appreciate the reviewer's diligent reading and valuable comments. Based on the suggestion, we have indeed removed the term "activator-free" from the abstract in the manuscript.

Reviewer 1, comment 2: On page 10, column 1, toward the bottom, the Authors assert that the failure of rxn to progress in the dark demonstrates that this is not a chain mechanism. Yoon and co-worker have shown, conclusively, that this kind of claim is erroneous since some chain processes terminate very quickly without irradiation. Please remove this claim. Here is Yoon's work which the Authors should read: <https://pubs.rsc.org/en/content/articlelanding/2015/sc/c5sc02185e>.

Response: We would like to express our gratitude to the reviewer for bringing this matter to our attention. As suggested, we have revised the statement "indicating not a light-initiated chain mechanism for the reaction" to "indicating that light played an important role in this reaction" on page 10, column 1, of the manuscript.

Reviewer 1, comment 3: Most importantly, in the SI, the Authors have failed to match the number

of significant figures in their measured quantities to the number of significant figures in the number of millimoles reported. In the preparation of 1c, for instance, the Authors indicate that 2.50 g (3 significant figures) of S2 correlates to 20.00 millimoles (4 significant figures). The number of significant figures should match. In the preparation of 1d, the Authors indicate that 0.81 g (2 significant figures) of S4 correlates to 5.09 millimoles (3 significant figures) of compound. The number of significant figures should match. At a glance, I can see dozens of mistakes like this throughout the SI. Please correct them. I will hold up the publication of this work for as long as is necessary until this problem is corrected.

Response: We greatly appreciate the diligent review conducted by the reviewer, and we would like to acknowledge that the errors that were brought to our attention were indeed oversights on our part. Consequently, we have carefully identified and rectified these errors in the Supplementary Information accordingly.

Reviewer 1, comment 4: Finally, I have noticed something that could be interesting in Figure 5. Note that the slope of the line going from 0-20 minutes is actually smaller than 40-60 minutes. It would appear that there is an induction period that one would not expect if the mechanistic hypothesis in the SI is correct. Is it possible, for instance, that free 2-hydroxytropolone forms an exciplex with the substrate donor and that this is more reactive than substrate donor alone? The Authors could consider adding 2-hydroxytropolone to their reaction mixtures to see if this enhances rate. While such an experiment is not something that I deem necessary for the current publication, it might enrich the work if the answer to my question is "yes."

Response: We would like to extend our gratitude to the reviewer for bringing this matter to our attention. In accordance with the reviewer's suggestion, we incorporated a catalytic amount of tropolone into the reaction mixture. Remarkably, we observed an acceleration in the reaction rate within the initial 20 minutes. The outcomes from these experiments have been included in Supplementary Table 8 and Supplementary Figure 6. Below are the specific details of this observation.

Supplementary Table 8. The effect of tropolone on the reaction of **1a** with **2a**.^[a]

Entry	Tropolone (equiv.)	Yield of 3a (%) ^b
1	0	4
2	0.1	8
3	0.3	11
4	0.5	12

[a] Reaction conditions: **1a** (0.075 mmol), **2a** (0.05 mmol), irradiation wavelengths, ClCH₂CH₂Cl (2.0 mL), argon atmosphere. [b] Isolated yields.

Supplementary Figure 6. The yield of reaction of **1a** with **2a** for 20 minutes

Reviewer #2:

In this revised manuscript, authors not only answered all my questions, but also include more useful

information along with addressing reviewers' comments. Thus, I recommend the manuscript publish in Nature Communications.

Thank you for your positive comments on our revised manuscript. We greatly appreciate your recommendation to publish our manuscript in Nature Communications.

REVIEWERS' COMMENTS

Reviewer #1 (Remarks to the Author):

I am satisfied with the Authors' revision. I recommend publication of this work in Nature Communications.